# The Short-Term Effect of Prunes in Improving Bone in Men

**DOI:** 10.3390/nu14020276

**Published:** 2022-01-10

**Authors:** Kelli S. George, Joseph Munoz, Lauren T. Ormsbee, Neda S. Akhavan, Elizabeth M. Foley, Shalom C. Siebert, Jeong-Su Kim, Robert C. Hickner, Bahram H. Arjmandi

**Affiliations:** 1Division of Animal and Nutritional Sciences, West Virginia University, Morgantown, WV 26506, USA; kelli.george@mail.wvu.edu; 2Center for Advancing Exercise and Nutrition Research on Aging, Florida State University, Tallahassee, FL 32304, USA; jm12t@my.fsu.edu (J.M.); lormsbee@fsu.edu (L.T.O.); nsa08@my.fsu.edu (N.S.A.); ss17u@my.fsu.edu (S.C.S.); jkim6@fsu.edu (J.-S.K.); 3Department of Nutrition, Food and Exercise Sciences, Florida State University, Tallahassee, FL 32304, USA; rhickner@fsu.edu; 4Department of Epidemiology and Cancer Control, St. Jude Children’s Research Hospital, Memphis, TN 38105, USA; elizabeth.foley@stjude.org; 5Institute for Successful Longevity, Florida State University, Tallahassee, FL 32304, USA

**Keywords:** osteoporosis, bone diseases, metabolic, men, inflammation, antioxidant

## Abstract

Osteoporosis is a major health concern in aging populations, where 54% of the U.S. population aged 50 and older have low bone mineral density (BMD). Increases in inflammation and oxidative stress play a major role in the development of osteoporosis. Men are at a greater risk of mortality due to osteoporosis-related fractures. Our earlier findings in rodent male and female models of osteoporosis, as well as postmenopausal women strongly suggest the efficacy of prunes (dried plum) in reducing inflammation and preventing/reversing bone loss. The objective of this study was to examine the effects of two doses of prunes, daily, on biomarkers of inflammation and bone metabolism in men with some degree of bone loss (BMD; t-score between −0.1 and −2.5 SD), for three months. Thirty-five men between the ages of 55 and 80 years were randomized into one of three groups: 100 g prunes, 50 g prunes, or control. Consumption of 100 g prunes led to a significant decrease in serum osteocalcin (*p* < 0.001). Consumption of 50 g prunes led to significant decreases in serum osteoprotegerin (OPG) (*p* = 0.003) and serum osteocalcin (*p* = 0.040), and an increase in the OPG:RANKL ratio (*p* = 0.041). Regular consumption of either 100 g or 50 g prunes for three months may positively affect bone turnover.

## 1. Introduction

Osteoporosis has significant public health importance for both women and men. It is estimated that 53.6 million adults have osteoporosis and low bone mineral density (BMD), representing approximately 54% of the adult population aged 50 years and older in the United States (U.S.) [1,2,3]. The culprit of many chronic diseases and conditions, including osteoporosis, is chronic inflammation [4]. It is known that inflammation alters bone remodeling, resulting in bone loss [5,6,7,8]. With time, this continual loss of bone may lead to impairments in the ability to perform everyday tasks, as well as an increased risk of falls and fractures [8,9].

Decreasing inflammation is an important therapeutic target for the prevention of osteoporosis. Inflammation is perpetuated by increases in oxidative stress [10,11]; therefore, antioxidants and substances rich in antioxidants, such as prunes (*Prunus domestica* L.), are of great interest for preventing the inflammatory state. Although there has been previous work that provides evidence for the effectiveness of prunes in preventing and reversing bone loss and inflammation in postmenopausal women, there is a lack of such studies in men.

Osteoporosis in men is an important yet understudied debilitating disease. One in four men over the age of 50 will succumb to an osteoporosis-related fracture, and men are more likely to die after an osteoporosis-related hip fracture than women [12]. Due to concerns regarding the increasing prevalence of male osteoporosis [12] and minimal treatment options available, it is imperative to further investigate the role of readily available preventative therapies, such as functional foods, on improving bone health in men. Previous research has demonstrated the bone-protective properties of prunes in male animal models of age-related bone loss, as well as gonadal hormone deficiency. In these studies, prunes were capable of both preventing bone loss, and more importantly reversing bone loss in male rats. Furthermore, a recent study [13] demonstrated that prunes were able to protect from bone loss caused by radiation in male mice. Overall, studies suggest that prunes are quite capable of reducing inflammation and preventing and reversing bone loss in both male [14,15] and female [16,17,18] rat models of osteoporosis, as well as improving bone biomarkers and bone mass in postmenopausal women [19,20]. Additionally, in one case study, a man who had substantial bone loss experienced a 7% gain in BMD after one year of consuming approximately 100 g of prunes plus calcium and vitamin D daily (unpublished data). This indicates that the beneficial findings regarding prunes in women, as is the case in many of the Food and Drug Administration (FDA) approved medications for osteoporosis, can perhaps be extrapolated to men. However, studies such as this clinical trial are needed to confirm this notion. If the findings of this study confirm prunes’ effectiveness in rebuilding bone or preventing further loss of BMD, this would indicate that prunes are an excellent, readily available, and natural treatment option.

In general, the consumption of fruits and vegetables has health-protective effects, in part, through the prevention of inflammation and its related conditions, including bone loss [21]. These health-protective effects have been mainly accredited to their bioactive components, including polyphenols [22,23]. Prunes have one of the highest oxygen radical absorbance capacities among fruits and vegetables consumed in the U.S. and are rich in a variety of nutrients, including vitamin K, magnesium, potassium, and boron [24], all of which play an important role in bone health.

Optimal bone health includes both high quantity of bone (bone density) and high quality of bone (bone microstructure). Previous clinical research on prunes and bone health has focused on bone density, as bone quality of the microstructure is more difficult to measure in humans. In this study, trabecular bone score (TBS), a newer endpoint measure of bone quality, was measured in addition to bone density to assess potential changes in bone quality that have previously not been evaluated in a clinical population given this intervention.

Based on the promising findings of previous research studies, the central hypothesis of the proposed study was that three months of regular prune consumption would reduce levels of inflammation, thereby improving biomarkers of bone health in men with some degree of bone loss. The secondary hypothesis of the study was that TBS would be negatively correlated with the assessed risk factors for osteoporosis.

## 2. Materials and Methods

### 2.1. Study Overview

For this present intervention study, 35 men with some degree of bone loss between the ages of 55 and 80 who were otherwise healthy were recruited from the greater Tallahassee, Florida area. Assessment for identification of potential subjects involved a short medical history and food and exercise questionnaires conducted over the phone. Potential subjects who were interested were next invited to the clinical research facility in the Sandels Building in the College of Human Sciences at Florida State University for the onsite screening. Potential subjects were provided with a verbal and written explanation of the informed consent during the initial (screening) visit. Detailed medical history, a food questionnaire, and physical activity levels of the potential subjects were obtained to confirm that the subject did not have any conditions violating the inclusion/exclusion criteria. If all the criteria were satisfied, anthropometric measurements (height, weight, and hip and waist circumference) were obtained, and then a licensed X-ray operator, under the supervision of a physician, assessed and analyzed the L1–L4 BMD using dual-energy X-ray absorptiometry (DXA). If subjects met all the inclusion criteria, they were assigned to treatment groups by using a pre-generated randomization list. Subjects were scheduled for an onsite clinical research Baseline and 3-month visit. Fasting venous blood draw, anthropometric measurements, urine sample, dietary recall, and physical activity questionnaires were performed at Baseline and 3-Month visits. Subjects were provided with their three-month supply of prunes, calcium, and vitamin D supplements, and calendars to monitor compliance at their baseline visit. The study was conducted according to the guidelines stated in the Declaration of Helsinki, and all procedures involving human subjects were approved by the Institutional Review Board (IRB00000446) at the Florida State University (Tallahassee, FL, USA). This clinical trial was registered at ClinicalTrials.gov: https://clinicaltrials.gov/ct2/show/NCT03408119 (accessed 12 December 2021).

### 2.2. Participant Inclusion Criteria

Healthy men between the ages of 55 and 80, whose lumbar (L1–L4) spine BMD t-score was between −0.1 and −2.5 SD below the mean were included. Participants who were not taking pharmacological agents known to affect bone, or who had not initiated an exercise program within the last six months known to influence bone were enrolled in the study. Subjects who met the inclusion criteria were considered for the study regardless of ethnicity or race.

### 2.3. Participant Exclusion Criteria

Men who had initiated a regular exercise regimen known to influence bone within the last 6 months prior to the study start date (e.g., resistance training) were excluded. Men whose BMD t-score at any site that was below 2.5 SD of the mean were excluded from the study. Subjects treated with aluminum antacids, anticonvulsants, calcitonin, bisphosphonates, Dilantin, denosumab, fluoride, neuroactive drugs, endocrine drugs, raloxifene, parathyroid hormone, anabolic agents, or steroids within a year prior to the start of the study were excluded. Subjects with metabolic bone disease, renal disease, cancer, cardiovascular disease, diabetes mellitus, respiratory disease, gastrointestinal disease, liver disease, severe or chronic asthma, eating disorders, or other chronic diseases were excluded. Those with a BMI < 18.5 and >40 kg/m^2^ were excluded to eliminate extremes in leanness and adiposity and to allow for body composition assessment. Subjects were additionally excluded if they smoked ≥20 cigarettes per day, or regularly consumed prunes or prune juice (≥3 servings/week). If subjects acquired any of these exclusion criteria at any point during the course of the study, they were disqualified.

### 2.4. Prescreening

Subjects were recruited through flyers, newspaper advertisements, and word of mouth throughout the greater Tallahassee, FL area. Interested subjects underwent an initial phone screening upon indicating interest in volunteering for the study. Basic informational and medical history questions were asked to assess eligibility for the study. Information on age, height, and weight to calculate BMI, presence of chronic disease or active cancer, medications, recent initiation of physical activity, current involvement in weight loss programs, smoking habits, and alcoholic beverage consumption habits were obtained over the phone. If the volunteer did not meet any exclusion criteria based on the phone screening questions, he was asked to come to the clinical research facility for a screening visit.

### 2.5. Screening

Research volunteers came to the clinical research facility at a time of their choosing for an in-person initial screening visit. During this visit, participants were provided with an informed consent form to sign at their will. Participants were advised that their participation in the study was completely voluntary, and a copy of the informed consent form was made available to them. Anthropometrics, including height, weight, and waist and hip circumference, were measured. Participants then underwent a DXA scan of the lumbar vertebrae (L1–L4) by a licensed operator to assess bone density. If the BMD of the lumbar spine (L1–L4) was between 0.1 and 2.5 SD below the mean, indicating mild bone loss, the participant was informed of their full qualification for the study. A detailed medical history, food intake questionnaire, and physical activity patterns were then collected to confirm prescreening findings and ensure participants did not meet any exclusion criteria. Upon qualification for the study, participants were scheduled for their baseline visit two weeks after the initial screening visit. They were given a three-day food record to take home with them and complete before their baseline visit.

### 2.6. Intervention

Qualified participants were randomly placed into one of three groups, determined by a pre-generated randomization list. Sample sizes per group are as follows: N = 15 in group A, N = 12 in group B, and N = 8 in group C. Group A consumed 100 g/day prunes for three months, group B consumed 50 g/day prunes for three months, and group C served as the control group. All three groups (groups A, B, and C) were also given one serving of a multivitamin containing 800 IU vitamin D and 450 mg calcium (dicalcium phosphate) to be taken daily alongside their treatment regimen for three months to provide a baseline protection against bone loss. Since the average calcium intake by American men exceeds 800 mg daily, this dose of 450 mg calcium was chosen to meet their Recommended Dietary Allowance (RDA) for calcium (1000 mg/day, ages 51–70; 1200 mg/day, ages 71+). The average intake of vitamin D_3_ for men from food alone is about 200 IU. Therefore, the addition of 800 IU supplemental vitamin D_3_ would meet their RDA requirements, aside from sun exposure. The rationale for choosing the doses of prunes was that both doses of prunes have previously positively influenced bone density and biomarkers of bone metabolism, and have been well-tolerated by postmenopausal women on a daily basis for 6 and 12 months [19]. In order to promote compliance, participants were informed of a variety of alternative ways to consume prunes (recipes available at: http://www.californiadriedplums.org/recipes, accessed 12 December 2021), since the polyphenols in prunes are heat resistant and will not be altered by cooking [25]. In addition, previous studies in which subjects did not gain body weight after consuming 100 g of prunes for one year, similar to the control group, indicate that prune supplementation is unlikely to disturb energy balance [19]. Prunes were donated in snack packs by the California Prune Board to be given to participants at no cost to them, and Shaklee Corporation provided the vitamin D and calcium containing multivitamin, again at no cost to participants.

### 2.7. Compliance

Compliance was monitored via the following means. First, the study participants were asked to return any remaining prunes and unused calcium and vitamin D_3_ supplements at their three-month final visit. Second, each individual was contacted via telephone on random dates to encourage compliance. Third, compliance to the study regimen was monitored via daily dosing calendars, which participants were instructed to return with them at their final visit. Participants were considered consistently compliant when meeting > 80% of their daily regimens.

### 2.8. Questionnaires

#### 2.8.1. Health and Medical History

A detailed health and medical history was obtained at baseline. The purpose of collecting this health and medical history data was to rule out men with acute or chronic conditions or diseases. The information collected also includes bone health history (i.e., incidence of previous fractures) and medication use.

#### 2.8.2. Dietary Assessment

The three-day food record was used to assess dietary intake. The Food Processor (ESHA Research, Salem, OR, USA) software was used to analyze dietary data. Each subject completed the three-day food record between their screening and baseline visits, and between their baseline and 3 month visits. Dietary data were analyzed to determine consistency and adequacy of dietary intake patterns over the study period and compliance with the study protocol.

#### 2.8.3. Physical Activity

Physical activity patterns were assessed at baseline and at the final visit. This study used the Five-City Project physical activity questionnaire, which has been validated to elicit information on current sleep and activity patterns [26]. It assesses total hours spent per week in leisure, occupational, and home activities and classifies them based on intensity. Physical activity data were analyzed to determine usual activity level, consistency over time, and deviations from baseline. If it was noticed that a participant initiated an exercise training program at any of the visits, he would have been excluded from the study.

### 2.9. Anthropometrics

Height was measured at baseline. Weight, waist, and hip circumference data were collected at baseline and at the final visit to monitor and suggest corrective measures should weight gain occur. The height of participants, after removal of their shoes, was measured using a wall-mounted stadiometer and their weight was measured using a digital scale (Seca Corporation, Chino, CA, USA). The waist and hip circumference of participants were measured using a Gulick fiberglass measuring tape with a tension handle (Creative Health Products, Inc., Ann Arbor, MI, USA). The waist circumference was measured in the horizonal plane midway between the lowest rib and the superior border of the right iliac crest. The hip circumference was measured around the widest portion of the buttocks with the measuring tape parallel to the ground. For measurement of both hip and waist circumference, participants stood with their feet together and arms by their sides while evenly distributing body weight. Measurements were repeated twice and if the measurements were within 1 cm of each other, the average of the two measurements was calculated. Study personnel ensured that the tape fit comfortably but did not compress the skin. 

### 2.10. Blood Collection and Processing

Fasting venous blood samples for plasma and serum analyses were collected between 6:00–10:00 a.m. on a designated date from each subject in vacutainers at baseline and 3-month visits. Blood samples were centrifuged at 3000× *g* for 15 min at 4 °C, aliquoted, and stored at −80 °C until analyses. 

### 2.11. Urine Collection

A spot urine sample after the first void was collected during the baseline and final visits. Urine was aliquoted from each collection and kept at −20 °C for future analyses. 

### 2.12. Bone Density

BMD analyses were conducted by a certified X-ray technologist. Measurements for each subject included total and regional body composition using the Lunar iDXA (GE Healthcare, Madison, WI, USA). DXA measurements were performed using the “compare analysis” option. Specific positioning and analysis guidelines were strictly followed according to the Lunar iDXA Operator’s Manual. Calibration, maintenance, and quality control of the iDXA was strictly maintained.

### 2.13. TBS

To obtain TBS, a newer endpoint measure of bone quality adopted by the FDA in 2012, lumbar spine DXA imaging was used to measure micro-architectural deterioration of bone, to aid as a predictor of fracture in men. TBS iNsight^®^ software (Medimaps, Chem. du Champ-des-Filles, Switzerland) was used alongside DXA images of the lumbar spine to obtain TBS, which assesses the DXA image texture. This novel measure of fracture risk was used to assess changes in bone quality in addition to the measure of bone quantity (density) from the DXA. TBS iNsight^®^ software (Medimaps, France) was used alongside DXA images of the lumbar spine for all screened participants (N = 90 for greater than 90% power). Trabecular bone score measures against a population average for age, and values range from low (0.8) to high (1.8), with higher values indicating higher bone microarchitecture, and therefore higher quality and lower risk of fracture [27].

### 2.14. Markers of Inflammation

To determine the potential role of systemic chronic inflammation in the bone-protective effects of prunes, serum levels of C-reactive protein (CRP) were assessed at baseline and at the final visit using commercially available ELISA kits (R&D Systems, Minneapolis, MN, USA) [28]. 

### 2.15. Markers of Antioxidant Activity

To determine the potential role of antioxidant activity in the bone-protective effects of prunes, serum levels of glutathione peroxidase (GPx) were measured at baseline and at the final visit using a Glutathione Peroxidase Assay Kit (Cayman Chemicals, Ann Arbor, MI, USA). The rationale for measuring GPx over superoxide dismutase is that GPx is produced, in part, by osteoblasts, to prevent cellular injury [29,30]. 

### 2.16. Markers of Bone Formation

One blood marker of bone formation, bone-specific alkaline phosphatase (BALP), was assessed in duplicate at baseline and at the final visit using commercially available ELISA kits (Quidel Biosystems, Mountain View, CA, USA). 

### 2.17. Markers of Bone Turnover

Three blood markers of bone turnover, osteocalcin, sclerostin and TRAP-5b, were assessed in duplicate at baseline and at the final visit using commercially available ELISA kits (Quidel Biosystems, Mountain View, CA, USA). These biomarkers were chosen due to previous research in postmenopausal women. 

### 2.18. Markers of Bone Health and Inflammation

Two blood markers of bone formation, osteoprotegerin (OPG) and receptor activator of nuclear factor kappa-B ligand (RANKL), were assessed in duplicate at baseline and at the final visit using commercially available ELISA kits (Quidel Biosystems, Mountain View, CA, USA and R&D Systems, Minneapolis, MN, USA, respectively). These biomarkers of bone formation were chosen to confirm previous findings in a male rat model of gonadal hormone deficiency and in postmenopausal women. 

### 2.19. Statistical Analysis

Statistical analyses were performed using SPSS software and the overall effects of the treatments were interpreted. Sample sizes were calculated using G*Power Software 3.1.9.2. Sample sizes were calculated based on estimated changes in CRP, as inflammation is a key factor in bone loss (primary outcome variable) [19]. The sample size of treatment groups required to achieve a power of 1 − β = 0.80 (group 1 mean of 1.75 ± 0.50 ng/mL; group 2 mean 1.24 ± 0.50 ng/mL) is 13. Descriptive statistics were calculated for all variables and include means, SD, medians, minima, and maxima. *p*-values < 0.05 were considered statistically significant. Distributions of outcome variables were examined graphically for symmetry and for outliers. If a lack of symmetry was observed, the variable was transformed before analysis. Any extreme outliers were examined for technical or clerical error. If the error was not attributed to technical or clerical errors, the data were included in analysis, and the result with deleting the outlying data point(s) was additionally reported. Baseline characteristics were compared between the study groups using one-way ANOVA. If significant differences were noted in variables that influence bone mass, such as physical activity levels, analyses were adjusted for the effects of these variables. This study was analyzed as a factorial design (3 treatments*2 timepoints) using repeated measures ANOVA. If a significant main effect was observed, post hoc analysis was run to locate significance. The relationship between TBS and risk factors for osteoporosis, including age, physical activity levels, body weight, and alcohol consumption within all groups and screened participants, were evaluated by Pearson’s correlation coefficient ^®^. The relationship between TBS and osteoporosis risk factors that are nominal variables, including ethnicity, family history, and smoking status were evaluated by the Eta statistic. The magnitude of correlation coefficients was considered as follows: very small to none (r < 0.1), small (0.1 ≤ r < 0.3), moderate (0.3 ≤ r < 0.5), large (0.5 ≤ r < 0.7), very large (0.7 ≤ r < 0.9), nearly perfect (0.9 ≤ r < 1.0), and perfect (r = 1.0).

## 3. Results

### 3.1. Baseline Characteristics

After screening 110 men, 44 men who met all inclusion criteria were randomly assigned to one of three treatment groups: Group A and B consuming 100 and 50 g/day prunes, respectively, for three months, and group C serving as the control group. Thirty-five men completed the study with group distribution as follows: group A, n = 15; group B, n = 12; group C, n = 8 (Figure 1). The most common reasons for attrition included self-reported non-compliance with the study protocol and personal reasons. Participants lost due to attrition were evenly distributed among treatment groups, and their baseline characteristics were not significantly different from those who completed the study. Baseline characteristics for the men who completed the study are presented in Table 1. Age, body weight, BMI, waist and hip circumference, and lumbar BMD (L1–L4) T-score did not significantly differ between groups at baseline.

### 3.2. Anthropometrics and Vitals

There were no significant changes from baseline both within and between groups in all anthropometrics measured, including weight, BMI, and waist-to-hip ratio (Table 2). There was a significant decrease in systolic blood pressure (SBP) in both the 50 g prune group (−6.1%, *p* = 0.042) and the control group (−6.7%, *p* = 0.003), but not in the 100 g prune group (Table 2). No significant changes were seen in diastolic blood pressure and pulse after 3 months, within or between groups in (Table 2).

### 3.3. Physical Activity

There were no significant changes in physical activity levels both within and between groups over the course of the study (Table 3).

### 3.4. Dietary Intake

Average intake of select nutrients from 3-day food records is reported for baseline and 3-month visits in Table 4. There was a significant group*time interaction for polyunsaturated fat intake (*p* = 0.029); however, multiple comparisons were not able to detect significant differences between groups. Intake of both monounsaturated fat and polyunsaturated fat significantly decreased within the 100 g prune group after 3 months (−30.2%, *p* = 0.001 and −31.4%, *p* = 0.032, respectively). Potassium intake was not significantly different between groups (*p* = 0.181); however, it was significantly greater after three months within the 100 g prune group (+23.7%, *p* = 0.043).

### 3.5. Serum Biomarkers of Inflammation and Bone Health

There were no significant changes in CRP (Figure 2) or GPx (Figure 3) both between and within groups from baseline to 3 months. There were no significant changes in BALP (Figure 4A), TRAP-5b (Figure 4C), or sclerostin (Figure 4B) both between and within groups from baseline to 3 months. There was a significant group*time interaction for OPG (*p* = 0.019) (Table 5), where there were significant decreases in OPG (Figure 4B) in the 50 g prune group and in the control group after three months (−38.9%, *p* = 0.003 and −53.3%, *p* = 0.004, respectively). There were no significant group*time interactions for the 50 or 100 g prune groups for RANKL (Figure 4C), but the control group showed a significant reduction in RANKL over the three months (−78%, *p* = 0.010). There was no significant difference between groups in osteocalcin levels (*p* = 0.105), but all three groups showed a significant decrease within groups (Figure 5A) over the three months (100 g prune, −33.2%, *p* < 0.001; 50 g prune, −14.8%, *p* = 0.040, control, −26.2%, *p* = 0.049). Furthermore, the OPG:RANKL ratio significantly increased in the 50 g prune group (*p* = 0.041).

### 3.6. Trabecular Bone Score

Results for the correlation between TBS and assessed osteoporosis risk factors are listed in Table 6. There was a significant negative correlation between age and TBS (r = −0.233, *p* = 0.013), and a significant positive correlation between body weight and TBS (r = 0.252, *p*= 0.007). There was no significant correlation between TBS and alcohol consumption, or physical activity levels. The correlations between TBS and ethnicity, and TBS and family history were considered small (Eta of 0.109 and 0.279, respectively), and the correlation between TBS and smoking status was considered moderate (Eta of 0.424).

## 4. Discussion

The bone-protective effects of prunes, including postulated mechanisms of action, have been outlined in our earlier review [31]. Previous research suggests that prunes are efficacious in both preventing and reversing bone loss via biomarkers of bone metabolism and inflammation in women, as indicated by both animal and clinical studies in postmenopausal women, and by improved BMD and trabecular structure in male rat models of osteoporosis. It has yet to be determined if these findings can be extrapolated to a male population. The findings of this study indicate that the short-term effects of daily prune consumption are minimal on inflammation and biomarkers of bone metabolism in men with mild bone loss. These findings are not consistent with findings from other investigators, though there is limited research in this population. Gaffen et al. [32] observed decreases in TRAP-5b and C-terminal collagen cross-links, markers of bone resorption, after three months of prune consumption by adult men. These investigators included all men regardless of bone density with a larger sample size of 66 men. Furthermore, our results indicate the TBS, as a measure of bone quality, is correlated with select risk factors for osteoporosis. Other research suggests TBS is consistently associated with fracture risk and prevalence but is not consistently associated with clinical risk factors for osteoporosis [33]. Simonelli et al. indicate TBS is negatively correlated with age [34]; however, it is accepted that TBS is at least partly independent of FRAX clinical risk factors [33], which include BMI, personal and family history of fracture, smoking status, steroid overuse, alcohol use, and presence of rheumatoid arthritis [35]. Our results indicated that in this population, TBS was negatively correlated with age and positively correlated with body weight. Furthermore, there was a moderate correlation between TBS and smoking status, where more nonsmokers had lower TBS scores (Table 5). Future studies in this population should additionally examine fracture risk to further increase knowledge on this newer measure of bone quality.

All three groups in the present study had similar baseline characteristics at the start of the study, which indicated adequate randomization. Dietary intake of potassium significantly increased in the 100 g prune group, indicating compliance with the study intervention. Interestingly, intake of both mono- and polyunsaturated fat significantly decreased in the 100 g prune group over the course of the study. The addition of prunes to the daily diet of participants may have replaced intake of sources of dietary fat, possibly explaining this decrease. Over the three months, the 50 g prune group and the control group had significant decreases in SBP, but the 100 g prune group saw no changes. This result is unexpected, but one possible explanation is that the control group had numerically greater SBP at baseline, leaving more room for improvement. Only one previous study on prunes reported significant changes in blood pressure with prune consumption [36], though there are limited studies that measured and reported changes in blood pressure.

Regarding bone activity, BALP, osteocalcin, OPG, TRAP-5b, sclerostin, and RANKL were measured before and after three months of prune consumption. In previous studies, three months of regular prune consumption significantly increased levels of BALP in postmenopausal women [37]; however, a recent study demonstrated decreases in BALP after three months of prune consumption in men [32]. In this study, there were no changes in BALP levels after three months. One year of prune consumption has previously effectively decreased levels of osteocalcin and sclerostin in postmenopausal women [20]. In this study, there were significant decreases in osteocalcin in all three groups, indicating a decrease in bone turnover, although the greatest percent decrease was seen in the 100 g prune group (−33%). Conversely, there were no changes seen in sclerostin after three months. Previously in male rats, prunes prevented the Orx-induced increase in RANKL expression, and one year of prune consumption numerically increased OPG production in postmenopausal women. Interestingly, in this study RANKL expression was only changed in the control group, where there was a significant decrease in expression. Since all treatment groups consumed the same amount of calcium and vitamin D-containing multivitamin, we have no treatment-related explanation for this result, and would suggest further study with a larger sample size. Of note, in this study we measured total RANKL levels, since the amount of RANKL circulating that is specifically bound to OPG is very small. RANKL can be found circulating either bound to OPG or in a free form, where the amount of the bound form is typically too small for commercially available ELISA kits to detect [38,39]. Furthermore, total RANKL expression was highly variable among all men in this study, which is consistent with other studies [40,41]. OPG values significantly decreased after three months in both the 50 g prune group and the control group, but not in the 100 g prune group, suggesting that consuming 100 g of prunes was sufficient to prevent the decrease in OPG, preventing a decrease in osteoblast activity. Interestingly, the OPG:RANKL ratio significantly increased in the 50 g prune group, but not in the 100 g prune group. A higher OPG:RANKL ratio indicates a greater amount of RANKL is bound to OPG, and not to osteoclasts, thus down-regulating osteoclast activity. In previous studies, both 100 and 50 g prune in postmenopausal women had no significant effect on the RANKL:OPG ratio [20,42]. OPG levels have been shown to be lower in men who reported drinking alcohol three or more days per week [43]. In this study population, 63% of the men had >1 drink per week, and 44% had >3 drinks per week. Though this does not explain the responsiveness to two different doses of prunes, it is possible that since many of our participants indicated they drink >3 days per week, their OPG activity may have had more room for improvement. Previous research has also indicated that women can have a greater baseline OPG:RANKL than men [44], indicating a possible gender difference in the ability to improve the ratio. Discrepancies in OPG and RANKL results could partly be due to using serum levels of RANKL and OPG, where other studies [14] have measured gene expression in bone. Furthermore, serum levels can be difficult to interpret with limitations of available ELISA kits [45,46].

One year of prune consumption has previously been shown to numerically decrease levels of sclerostin, albeit not significantly, and both 50 and 100 g of daily prune consumption for six months was able to significantly decrease TRAP-5b production, which is indicative of osteoclast activity, in both postmenopausal women and men [32,42]. In this study, no changes were observed in either sclerostin or TRAP-5b levels after three months.

Some limitations to the current study include the sample size, as a larger sample size would achieve greater power to observe the short-term effects of prune consumption on bone health and inflammation. However, this is a subset of a larger study (results not yet published). The participants recruited were also generally healthy (no severe bone loss, free from common chronic diseases, such as type II diabetes), so the results may also only be generalized to a similar population. The supplement used to provide participants with calcium and vitamin D as baseline protection against bone loss was a multivitamin, rather than individual calcium and vitamin D supplements, which may have its own effect on bone. This multivitamin did contain other nutrients, including antioxidants and bone-promoting agents, such as vitamin E, phosphorous, magnesium, copper, and boron, which could have influenced positive results seen in the control group. Future studies should isolate calcium and vitamin D for baseline protection against bone loss. Additionally, our population, though randomly selected, was not ethnically diverse, with most of the sample being Caucasian. The study population was also recruited from a region with larger amounts of yearly sunlight, therefore may have greater baseline exposure to vitamin D. Lastly, testosterone levels were not assessed through the course of the study, which are known to be associated with lower bone density in men and may have had an influence on bone health in this population.

## 5. Conclusions

Previous clinical research has demonstrated the beneficial effects of short- and long-term consumption of prunes in maintaining and preventing the loss of bone density, as well as decreasing inflammation, in postmenopausal women. Though the aim of this study was to examine whether similar effects could be seen in men, the results of this study indicate that after three months of regular consumption, prunes have a minimal effect on levels of inflammation, though modest improvements were seen in some bone biomarkers. Three months of consumption may not be long enough to manifest changes in bone metabolism; therefore, larger-scale studies looking at the short- and long-term effects of prunes on bone health and inflammation are warranted. Furthermore, TBS may be a useful tool in evaluating bone quality in similar populations, as it was correlated with a number of risk factors for osteoporosis. Future studies should also examine the long-term effects of prunes on TBS to examine any potential change in bone quality over a longer period of time.

## Figures and Tables

**Figure 1 nutrients-14-00276-f001:**
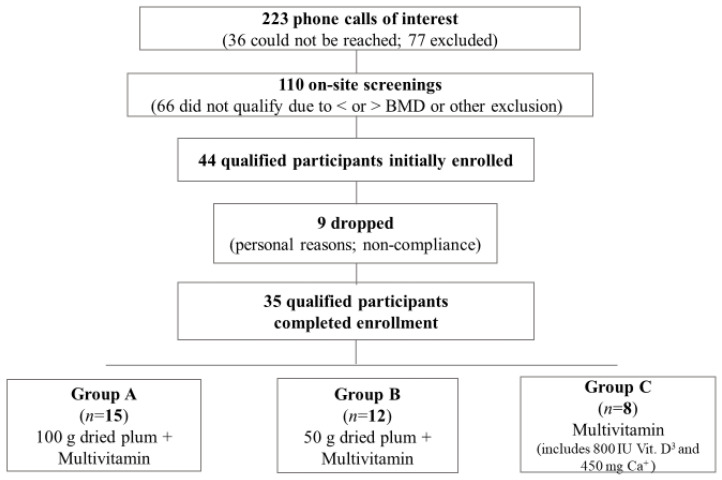
Study Flowchart. Abbreviations: Bone Mineral Density, BMD.

**Figure 2 nutrients-14-00276-f002:**
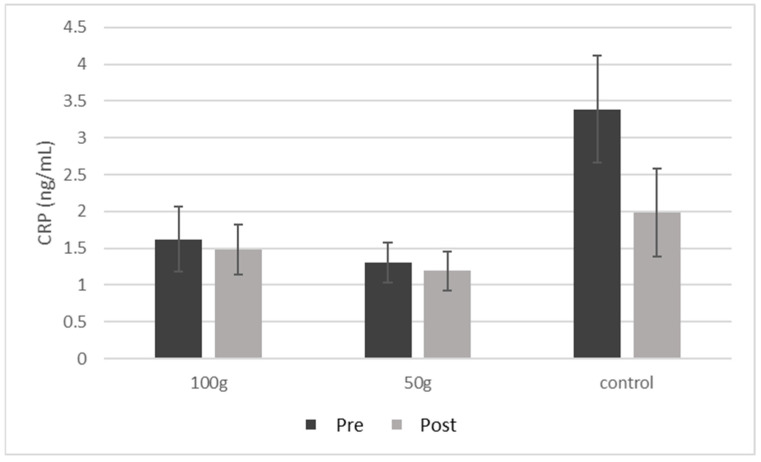
Blood C-reactive protein in men before and after three months of prune consumption as a measure of inflammation. Bars represent mean ± SEM.

**Figure 3 nutrients-14-00276-f003:**
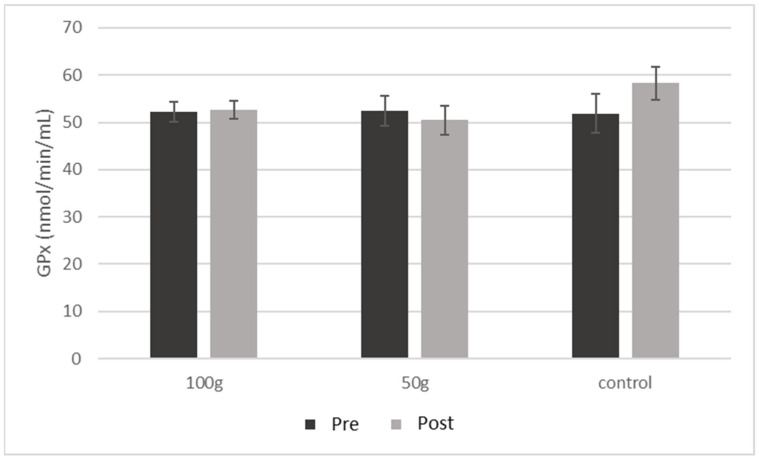
Blood glutathione peroxidase in men before and after three months of prune consumption as a measure of antioxidant activity. Bars represent mean ± SEM.

**Figure 4 nutrients-14-00276-f004:**
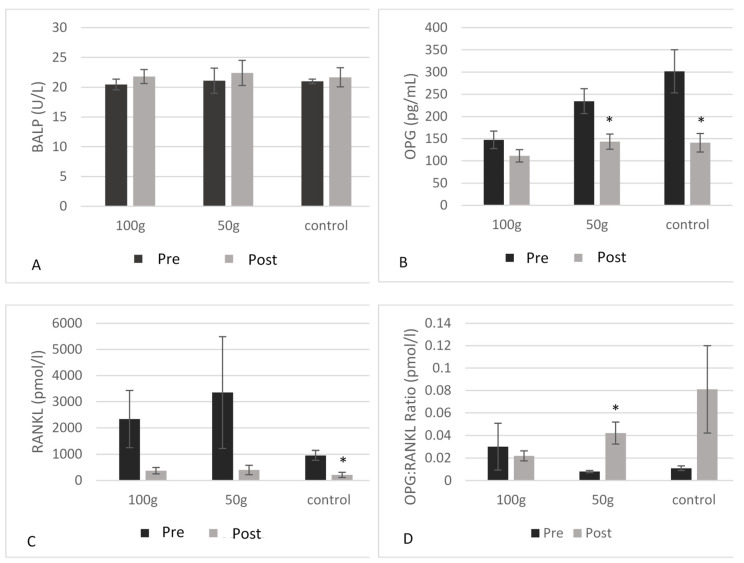
Blood BALP (**A**), blood OPG (**B**), blood RANKL (**C**), and the OPG:RANKL ratio (**D**) in men before and after three months of prune consumption as measures of bone formation. Bars represent mean ± SEM. * denotes significant (*p* < 0.05) difference compared to baseline.

**Figure 5 nutrients-14-00276-f005:**
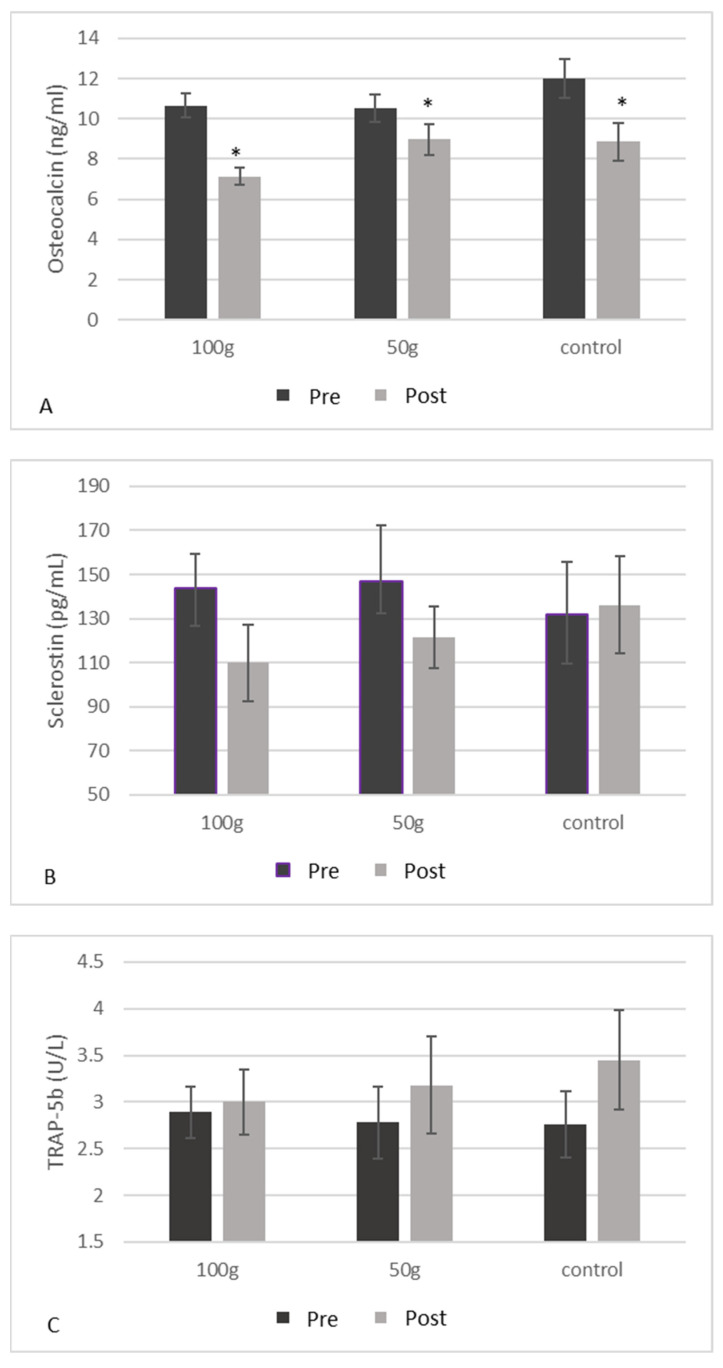
Blood osteocalcin (**A**), blood sclerostin (**B**), and blood TRAP-5b (**C**) in men before and after three months of prune consumption as measures of bone turnover. Bars represent mean ± SEM. * denotes significant (*p* < 0.05) difference compared to baseline.

**Table 1 nutrients-14-00276-t001:** Baseline characteristics between and among groups.

Measures	100 g DP	50 g DP	Control	*p* Value
Age (years)	65.9 ± 7.3	69.1 ± 5.9	64.9 ± 5.9	0.316
Height (cm)	173.8 ± 4.4	175.1 ± 9.0	174.6 ± 10.3	0.914
Weight (kg)	74.7 ± 13.7	82.1 ± 13.7	87.6 ± 24.2	0.206
BMI (kg/m^2^)	24.6 ± 3.7	24.3 ± 8.7	28.6 ± 6.9	0.312
WC (cm)	89.8 ± 13.4	95.7 ± 12.6	99.4 ± 15.3	0.296
HC (cm)	97.1 ± 7.3	106.7 ± 9.7	103.7 ± 12.3	0.063
BMD T-score (L1–L4)	−1.01 ± 1.21	−0.60 ± 0.93	−0.78 ± 0.67	0.678

Abbreviations: Prunes, DP; Standard Deviation, SD; Body Mass Index, BMI; Waist Circumference, WC; Hip Circumference, HC; Bone Mineral Density, BMD.

**Table 2 nutrients-14-00276-t002:** Anthropometric and vital measurements at baseline (Pre-) and 3 months (Post).

	100 g DP (mean ±SD)			50 g DP (mean ±SD)			Control (mean ±SD)			Trt*Time
Measures	Pre	Post	*p* Value	Pre	Post	*p* Value	Pre	Post	*p* Value	*p* Value
Weight (kg)	75.9 ± 13.4	75.5 ± 13.6	0.146	82.1 ± 13.7	81.8 ± 13.0	0.540	86.5 ± 25.9	86.4 ± 25.8	0.863	0.438
BMI (kg/m^2^)	25.0 ± 3.5	24.9 ± 3.5	0.132	26.5 ± 4.3	26.4 ± 4.0	0.538	28.6± 2.5	28.7± 2.4	0.540	0.415
WHR	0.93 ± 0.08	0.92 ± 0.09	0.473	0.90 ± 0.08	0.91 ± 0.05	0.344	0.96 ± 0.06	0.95± 0.07	0.648	0.420
SBP (mmHg)	133.2 ± 23.2	130.8 ± 18.0	0.432	138.3± 16.3	129.8 ± 12.2	0.042 *	149.7 ± 18.3	139.4 ± 13.5	0.003*	0.171
DBP (mmHg)	76.9 ± 12.8	76.3 ± 11.8	0.834	77.6 ± 13.4	75.7 ± 9.3	0.448	81.3 ± 11.2	80.1 ±12.7	0.565	0.920
Pulse (bpm)	62.1 ± 7.3	62.0 ± 6.6	0.982	67.6 ± 16.1	64.3 ± 13.9	0.186	66.7± 9.6	68.9 ± 7.4	0.204	0.222

Abbreviations: Prunes, DP; Standard Deviation, SD; Body Mass Index, BMI; Waist-to-Hip Ratio, WHR; Systolic Blood Pressure, SBP; Diastolic Blood Pressure, DBP. * denotes significance (*p* < 0.05).

**Table 3 nutrients-14-00276-t003:** Physical activity levels from baseline (Pre-) to 3 months (Post).

	100 g DP			50 g DP			Control			Trt*Time
Measures	Pre	Post	*p* Value	Pre	Post	*p* Value	Pre	Post	*p* Value	*p* Value
Total PA (hrs/week)	13.0 ± 14.3	13.1 ± 13.1	0.995	9 ± 7.0	15.2 ± 11.9	0.058	4.6 ± 4.9	3.8 ± 5.6	0.526	0.274

Abbreviations: Prunes, DP; Standard Deviation, SD; Physical Activity, PA. * denotes significance (*p* < 0.05).

**Table 4 nutrients-14-00276-t004:** Average dietary intake from baseline (Pre-) to 3 months (Post).

	100 g DP (mean ± SD)			50 g DP (mean ± SD)			Control (mean ± SD)			Trt*Time
Measures	Pre	Post	*p* Value	Pre	Post	*p* Value	Pre	Post	*p* Value	*p* Value
Total kcal	1920 ± 400	1927 ± 682	0.964	1590 ± 459	1617 ± 450	0.853	1647 ± 157	1637 ± 403	0.953	0.989
Protein (g)	87.1 ± 26.8	92.7 ± 33.5	0.461	77.8 ± 23.6	92.3 ± 61.6	0.523	69.9 ± 13.4	62.7 ± 9.5	0.253	0.745
CHO (g)	214.2 ± 62.1	256.1 ± 85.8	0.062	188.2 ± 86.4	173.9 ± 80.0	0.603	222.0 ± 47.4	231.5 ± 66.6	0.807	0.265
Total Fiber (g)	26.8 ± 13.1	30.4 ± 13.1	0.230	17.6 ± 7.7	19.7 ± 6.9	0.099	22.8 ± 9.9	20.5 ± 11.6	0.408	0.543
Sugar (g)	82.7 ± 24.1	91.4 ± 41.6	0.450	70.8 ± 43.5	78.9 ± 33.5	0.410	68.0 ± 17.4	89.9 ± 53.1	0.358	0.759
Total Fat (g)	73.9 ± 20.2	67.1 ± 29.3	0.409	56.2 ± 20.2	65.4 ± 31.7	0.362	53.2 ± 7.2	51.8 ± 13.3	0.801	0.401
Saturated Fat (g)	22.2 ± 8.2	18.7 ± 9.4	0.070	18.3 ± 7.5	25.6 ± 27.5	0.398	16.8 ± 3.0	15.2 ± 2.6	0.516	0.407
MUFA (g)	16.9 ± 7.1	11.8 ± 7.0	0.001 *	11.9 ± 8.1	13.5 ± 7.0	0.579	8.5 ± 2.0	8.5 ± 1.4	0.987	0.091
PUFA (g)	10.5 ± 3.7	7.2 ± 3.4	0.032 *	6.6 ± 3.6	10.4 ± 7.9	0.119	8.3 ± 2.8	7.6 ± 3.4	0.675	0.029*
Trans Fat (g)	0.18 ± 0.34	0.22 ± 0.23	0.651	0.20 ± 0.22	3.0 ± 8.5	0.323	0.13 ± 0.16	0.51 ± 0.63	0.201	0.548
Cholesterol (mg)	284.3 ± 160.6	254.0 ± 151.6	0.442	317.3 ± 178.6	261.6 ± 150.4	0.322	207.6 ± 101.4	181.4 ± 88.2	0.688	0.903
Potassium (mg)	1791 ± 489	2215 ± 772	0.043 *	1800 ± 634	1724 ± 601	0.730	1626 ± 354	1439 ± 639	0.693	0.181
Sodium (mg)	2328 ± 732	2536 ± 1327	0.553	2494 ± 1117	2294 ± 1574	0.636	1802 ± 408	2108 ± 467	0.226	0.632

Abbreviations: Standard Deviation, SD; Kilocalories, kcal; carbohydrates, CHO; Monounsaturated fat, MUFA; Polyunsaturated fat, PUFA. * denotes significance (*p* < 0.05).

**Table 5 nutrients-14-00276-t005:** Biomarkers of inflammation and bone health from baseline (Pre-) to 3 months (Post).

	100 g DP (mean ± SD)			50 g DP (mean ± SD)			Control (mean ± SD)			Treatment*Time
Measure	Pre	Post	*p* Value	Pre	Post	*p* Value	Pre	Post	*p* Value	*p* Value
CRP (ng/mL)	1.62 ± 1.6	1.48 ± 1.2	0.188	1.30 ± 0.8	1.19 ± 0.8	0.252	3.39 ± 1.6	1.98 ± 1.3	0.216	0.171
GPx (nmol/min/mL)	52.31 ± 8.2	52.66 ± 7.7	0.919	52.44 ± 10.5	50.43 ± 10.0	0.699	51.81 ± 10.9	58.27 ± 9.2	0.419	0.545
OPG (pg/mL)	147.2 ± 80.99	111.4 ± 57.87	0.155	234.5 ± 92.84	143.2 ± 56.60	0.003 *	301.6 ± 128.4	140.9 ± 55.20	0.004 *	0.019 *
RANKL (pmol/L)	2337 ± 2885	373.6 ± 330.7	0.129	3347 ± 4776	403.0 ± 398.9	0.208	955.2 ± 420.1	210.4 ± 222.6	0.010 *	0.535
OPG: RANKL ratio (pmol/l)	0.030 ± 0.051	0.022 ± 0.011	0.685	0.008 ± 0.002	0.042 ± 0.020	0.041 *	0.011 ± 0.004	0.081 ± 0.069	0.217	0.029 *
OC (ng/mL)	10.67 ± 2.39	7.13 ± 1.69	<0.001 *	10.54 ± 2.25	8.98 ± 2.58	0.040 *	12.01 ± 2.55	8.86 ± 2.45	0.049 *	0.105
Sclerostin (pg/mL)	143.8 ± 58.5	110.0 ± 62.7	0.088	146.7 ± 80.9	121.5 ± 44.8	0.460	131.7 ± 63.1	136.1 ± 58.3	0.839	0.543
BALP (U/L)	20.47 ± 3.4	21.80 ± 4.3	0.141	21.10 ± 5.5	22.4 ± 5.5	0.300	20.99 ± 3.7	21.67 ± 4.3	0.516	0.289
TRAP-5b (U/L)	2.89 ± 1.12	3.00 ± 1.39	0.651	2.78 ± 1.28	3.18 ± 1.71	0.362	2.76 ± 0.91	3.45 ± 1.41	0.284	0.135

Abbreviations: Dried Plums, DP; C-Reactive Protein, CRP; Glutathione Peroxidase, GPx; Osteoprotegerin, OPG; Osteocalcin, OC; Bone-Specific Alkaline Phosphatase, BALP; Receptor Activator of Nuclear Factor Kappa-B Ligand, RANKL; Tartrate-Resistant Acid Phosphatase, TRAP-5b. * denotes significance (*p* < 0.05).

**Table 6 nutrients-14-00276-t006:** Correlation between selected osteoporosis risk factors and trabecular bone score in all participants at baseline.

Risk Factor	Correlation (r)	Significance (*p* Value)
Age	−0.233	0.013 *
Ethnicity ^a^	0.109	N/A
Physical activity levels	−0.031	0.866
Body weight	0.252	0.007 *
Family history ^a^	0.279	N/A
Smoking status ^a^	0.424	N/A
Alcohol consumption	−0.063	0.694

^a^ For these variables, eta value is used in lieu of Pearson’s correlation coefficient. * denotes significant correlation (*p* < 0.05).

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
