# Peer review of "The Short-Term Effect of Prunes in Improving Bone in Men"

_nutrients, 2022, doi:10.3390/nu14020276_

Round 1
Reviewer 1 Report
This manuscript reported a randomized control trial on the short-term effect of prune in improving bone in men. This was a very interesting study.
- The manuscript gave a very through description of the design of the study. The sample size calculation was based on the changes of CRP. Was this the primary outcome of the study? The assumed change was 1.75 ±50 ng/mL for group one. Was there a reference for this change? Based on Figure 1, the changes was very little in groups 1 and 2.
- Were two-sample t-tests used in pre-post comparisons (Table 2-4)? Why was difference not used, meaning pre minus post for each individual? Were all three groups combined for the correlation in Table 5?
3. The significance reported in Figure 4 as well as in abstract was based
on two groups comparisons. This might subject to multiple
comparisons. Adjustment was required.
Author Response
Response to Reviewer 1
We thank the reviewer for his/her feedback and suggestions.
- CRP was the primary outcome of the study. This has now been clarified in the statistical analysis section, and a reference has been added. These numbers came from both a previous study in women (Hooshmand, 2010 – referenced in paper), as well as g*power analysis.
- The title of Table 5 has been edited to clarify those that were included in this analysis. This is also mentioned in the methods section (line 261).
We agree with the reviewer that these statistics have been done differently, but we were advised by our statistician to run the analysis using mean values. Additionally, previous similar studies in women by our laboratory were published in a similar manner, again as advised by our statistician.
- Thank you for your comment. The authors had previously run multiple groups comparisons, but excluded the raw data in table format, inputting the p values in the abstract and text of the results, as the reviewer noticed. A table with this data has been added as Table 6 in the text to clarify this.
Reviewer 2 Report
General comments
- I found the study well designed, the paper well written, and the research question useful.
- Consider adding some discussion of the limited generalizability of this study. Because of the exclusion criteria, the findings are limited to a population in pretty good health in terms of both bone density and common chronic diseases like heart disease and diabetes. Recruitment from a sunny region like Florida would likely also mean higher baseline vitamin D levels and perhaps slightly more physical activity in the study population, compared to a wintry climate.
Specific comments
Title – make “prunes” plural
Line 22 – “men with some degree of bone loss”
Line 90 – “interventional” study
Line 102 – define DXA at first use
Line 127 – spell out PTH
Line 202 – “food record was used…”
Line 218 – “corrective measures”
Line 220 – “measures” should be “measured”
Line 225 – “potion” should be “portion”
Line 229 – “within 1 cm”
Line 247 – Section 2.13 – What is the possible range of TBS, and what range is considered normal? Is it measured against a population average, like BMD, or something else? Is a higher score better or worse? It is important to describe these details so that the direction of correlations (positive or negative) with osteoporosis risk factors in section 3.6 can be readily interpreted.
Line 248 – FDA is already abbreviated on line 64
Lines 259, 263 – make “prunes” plural
Line 259 – Define CRP at first use
Line 263 – define GPx at first use
Line 265 – spell out SOD
Line 267 – define BALP at first use
Line 276 – define OPG at first use, and RANKL if appropriate
Line 295 – “variables that influence bone mass”, such as? Give some examples in parentheses.
Line 296-297 – Check that this sentence is written correctly. The repetition of “repeated measures” sounds a little odd.
Lines 298-299 – Check that the beginning of this sentence is written correctly. The relationships being checked for correlation are not entirely clear.
Line 303 – Change to “magnitude…was”
Lines 310-311 – “…g/day prunes, respectively…”
Line 313 – remove “lost”
Line 318 – lumbar
Line 323 – define SBP at first use
Line 333 – Why are some values in italics in Table 2?
Line 336 – Heading for section 3.3 is repeated here and line 327.
Line 339 – For Table 3, it would be helpful to describe the numerical scale of the questionnaire in the Methods text, so readers will understand the numbers here.
Line 343 – “average intake…is reported…”
Line 344 – capitalize “There”
Line 364 – spell out osteocalcin, here and elsewhere (lines 441-442)
Line 375 – panel D is missing its label
Line 432 – remove “have”
Line 461 – change “number” to “amount”
Line 466 – change “drink” to “had” in both instances
Author Response
Response to Reviewer 2
We thank the reviewer for his/her detailed feedback and suggestions. All edits to the document text have been made according to the reviewer’s input.
Additionally, a discussion of the generalizability of the results has been added to the discussion of other limitations (lines 519 and 529), and a further explanation of TBS values has been added to the methods section (line 262).
Lastly, a further description of the numerical scale for the physical activity questionnaire has been added to the methods section as well (line 216).